# V2M: Visual 2-Dimensional Mamba for Image Representation Learning

## Abstract

Mamba has garnered widespread attention due to its flexible design and efficient hardware performance to process 1D sequences based on the state space model (SSM). Recent studies have attempted to apply Mamba to the visual domain by flattening 2D images into patches and then regarding them as a 1D sequence. To compensate for the 2D structure information loss (e.g., local similarity) of the original image, most existing methods focus on designing different orders to sequentially process the tokens, which could only alleviate this issue to some extent. In this paper, we propose a Visual 2-Dimensional Mamba (V2M) model as a complete solution, which directly processes image tokens in the 2D space. We first generalize SSM to the 2-dimensional space which generates the next state considering two adjacent states on both dimensions (e.g., columns and rows). We then construct our V2M based on the 2-dimensional SSM formulation and incorporate Mamba to achieve hardware-efficient parallel processing. The proposed V2M effectively incorporates the 2D locality prior yet inherits the efficiency and input-dependent scalability of Mamba. Extensive experimental results on ImageNet classification and downstream visual tasks including object detection and instance segmentation on COCO and semantic segmentation on ADE20K demonstrate the effectiveness of our V2M compared with other visual backbones.

## 1 Introduction

State Space Models (SSM) have recently received sustained attention in natural language processing (NLP), achieving excellent performance on various tasks because of its efficiency in handling long sequences (Gu et al., 2021a; 2022; 2021b). Mamba (Gu & Dao, 2024) further equip the conventional SSMs with the ability to process time-varying input and exquisitely designed hardware acceleration algorithms, demonstrating the potential to compete with transformers (Kim et al., 2018).

Although convolutional neural networks (CNNs) (He et al., 2016; Liu et al., 2022) and visual transformers (ViTs) (Dosovitskiy et al., 2020; Liu et al., 2021) have consistently dominated computer vision, recent works applying Mamba to visual data modeling emerges as promising alternatives (Zhu et al., 2024; Liu et al., 2024b; Huang et al., 2024). To adapt 2D visual data to Mamba with 1D sequential processing, most existing methods follow ViTs to patchify images into tokens and design various ordering strategies to incorporate 2D structural prior. For example, Vision Mamba (Vim) (Zhu et al., 2024) directly flattens images by rows and then employs a bidirectional modeling strategy to enhance the latent representation of the model. LocalMamba (Huang et al., 2024) attempts to preserve the local invariance of images through local scanning and optimal scanning direction search. However, these works are still constrained in the framework of 1-dimensional Mamba, which can only approximate the 2D structural modeling of images and inevitably disrupts the coherence and locality of visual data.

In this paper, we propose a Visual 2-Dimensional Mamba (V2M) model as a complete solution for 2D visual modeling. We directly processes image tokens in the 2D space instead of transforming them into 1D sequences to adapt to the 1D Mamba, as illustrated in Figure 1. Specifically, we first generalize the state space model to the 2-dimensional space which generates the next state considering two adjacent states on both dimensions (e.g., columns and rows). We then construct our V2M based on the 2-dimensional SSM formulation and incorporate Mamba to achieve hardware-efficient parallel processing. In addition, considering the non-sequential nature of image tokens, we construct

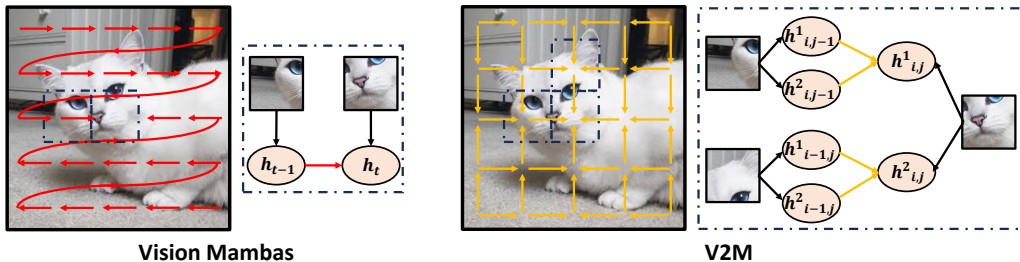

Figure 1: Motivation of the proposed V2M method. Previous vision Mambas processed image tokens with 1D SSM, whereas we extend SSM to a 2D form for more suitable image representation learning by introducing the prior of enhancing the relevance of adjacent regions for modeling. ($h^1_{i,j}$ denotes the horizontal state at position (i,j) while $h^2_{i,j}$ is the vertical state at the same position.)

the 2-dimensional state equations starting from all four corners of the image. Our proposed V2M approach essentially preserves the local similarity and coherence of image representations, and also inherits the efficiency and input-dependent scalability of Mamba. We conduct extensive experiments on the ImageNet-1K dataset for classification, similar to previous settings of convolution networks and vision transformers. Furthermore, we transfer the pretrained models to downstream tasks such as object detection, instance segmentation, and semantic segmentation to evaluate the transferability of V2M. We report both the ImageNet classification accuracy and the corresponding metrics of the downstream tasks for comparison. Experimental results demonstrate the consistent improvements of our proposed method compared with the Vim baseline, verifying the superiority of V2M (e.g. $+0.4\%$ for Vim on ImageNet-1K).

## 2 RELATED WORK

**Generic Vision Backbones.** Convolutional Neural Networks (CNNs) and Vision Transformers (ViTs) are the most widely adopted generic backbones in computer vision. Among them, CNNs leverage shift-invariance for feature extraction for each layer, setting a series of new benchmarks on numerous visual tasks (He et al., 2016; Liu et al., 2022; Krizhevsky et al., 2012; LeCun et al., 1998; Szegedy et al., 2015; Simonyan & Zisserman, 2014; Huang et al., 2017). Nevertheless, the introduction of ViT has changed the landscape of computer vision, shifting the dominance of CNNs as the backbone to a new paradigm (Dosovitskiy et al., 2020; Liu et al., 2021; Touvron et al., 2021; Carion et al., 2020; Caron et al., 2021). Specifically, the plain ViT (Chen et al., 2020) divides an image into multiple patches and linearly embeds these patches as the input sequence for the Transformer model with positional encoding. Swin Transformer (Liu et al., 2021) introduces hierarchical feature maps and a self-attention mechanism within local windows, which enables the model to effectively handle image features at different scales. Transformer-based models are also widely employed in multimodal tasks due to their versatility in handling data from different modalities.

In addition to CNNs and ViTs, a plethora of sophisticated visual architectures based on Mamba (Gu & Dao, 2024), have recently emerged in the domain of computer vision (Zhu et al., 2024; Liu et al., 2024b; Huang et al., 2024; Liu et al., 2024a; Ruan & Xiang, 2024; Yang et al., 2024; Pei et al., 2024). For example, Vision Mamba (Vim) (Zhu et al., 2024) directly transfers Mamba from NLP to the visual domain, which does not incorporate significant considerations specifically for visual data. VMamba (Liu et al., 2024b) addresses these deficiencies through further exploration, utilizing one-dimensional scans in four distinct directions to simulate the modeling of two-dimensional images, and adopting a hierarchical structure for feature extraction. LocalMamba (Huang et al., 2024) further divides the input image into multiple local windows, thereby retaining the local similarity of the image to a certain extent. Additionally, PlainMamba (Yang et al., 2024) is designed with a non-hierarchical structure to enhance feature fusion and modality fusion, whereas EfficientV-Mamba (Pei et al., 2024) conducts a streamlined and detailed study of the lightweight adaptation of VMamba. These endeavors harness the high computational and memory efficiency of Mamba, yet predominantly involve the naive flattening of images input for the original 1-dimensional Mamba formulation, which compromises the local similarity and coherence of images and thus results in

suboptimality. On the contrary, we employ 2-dimensional state space equations to model image features, and thereby effectively incorporates the 2D locality prior, establishing a more rational application of Mamba within the realm of computer vision.

**State Space Models.** State space models (SSMs) represent the output of the system (observation) as a function of the state of the system and typically assume that the evolution of the system state over time follows a Hidden Markov Model (HMM). Considering the proficiency in the modeling of extended sequences, 1-dimensional SSMs have garnered extensive and conspicuous attention within the domain of natural language processing (Gu et al., 2022; 2021a;b; Gupta et al., 2022; Gu & Dao, 2024). For example, S4 (Gu et al., 2021a) transforms the SSM into a large 1D convolution to achieve efficient parallelization and adopts HiPPO matrices as initialization to facilitate memory storage and prevent gradient explosion or vanishing. S4D (Gu et al., 2022) systematically investigates the parameterization and initialization methods for diagonal state space models. Mamba (Gu & Dao, 2024) further improves the traditional state space model, enabling it to selectively extract information based on the input content, combined with a hardware-aware parallel algorithm that achieves efficient processing.

Several studies have attempted to extend the formulation of the state space models into a 2-dimensional structure (Kung et al., 1977; Eising, 1978; Kurek, 1985; Hinamoto, 1980). They typically increase the state dimension in the state space model and perform corresponding state iterative calculations along each dimension. In computer vision, Baron *et al.* (Baron et al., 2023) integrates a 2-dimensional state space model into CNN and ViT architectures, implementing it as an additional layer for the processing of features. Nonetheless, this approach neglects the input-dependent characteristic of SSM, which hampers the capacity of the model for adaptively extracting features from the input images. In contrast, our proposed V2M generalizes the SSM to the 2-dimensional space which generates the next state considering two adjacent states on both dimensions, while also preserving the input-relevant feature of the SSM.

**Visual Representation Learning.** Visual representation learning aims to acquire features that are capable of encapsulating and portraying the essence of visual data, which are widely harnessed in a diverse array of visual tasks including image classification, object detection, image segmentation, pose estimation, and video analysis. Typical settings include pre-training on large datasets (such as ImageNet) and then fine-tuning on downstream tasks and specific datasets (He et al., 2020; Grill et al., 2020; He et al., 2022; Zhou et al., 2022; Chen & He, 2021; Khosla et al., 2020; He et al., 2016; Liu et al., 2021), which can be divided into supervised settings (Khosla et al., 2020; He et al., 2016; Liu et al., 2021) and unsupervised patterns (He et al., 2020; Grill et al., 2020; He et al., 2022; Zhou et al., 2022; Chen & He, 2021). Specifically, supervised representation learning utilizes the ground truth labels of samples as the supervision while unsupervised representation learning constructs a label-free pretext task to conduct the training procedure, such as contrastive learning (He et al., 2020; Grill et al., 2020; Chen & He, 2021) and mask image modeling (He et al., 2022; Zhou et al., 2022). In this paper, we follow the setting of supervised representation learning. We pretrain the model with ground truth labels on the image classification task and then transfer to object detection, instance segmentation, as well as instance segmentation for verification of transferability.

## 3 PROPOSED APPROACH

In this section, we first present general preliminaries of 1-dimensional state space models, Mamba, and 2-dimensional state space models. Then we elaborate on the implementations of the 2-dimensional state space models on visual representation learning. Lastly, we provide an overview of the proposed V2M framework.

### 3.1 STATE SPACE MODELS FOR VISION

The state space models (SSMs) serve as the foundation for Mamba. However, conventional state space models exhibit constrained proficiency in the extraction of features from input data, an outcome attributed to the time-invariant nature of the parameters within the equations. Mamba rectifies this limitation by integrating time-varying parameters. Concretely, Mamba formulates the projection parameters ($\mathbf{B} \in \mathbb{R}^{B \times L \times N}$, $\mathbf{C} \in \mathbb{R}^{B \times L \times N}$) and timescale parameter $\mathbf{\Delta} \in \mathbb{R}^{B \times L \times D}$ in a manner

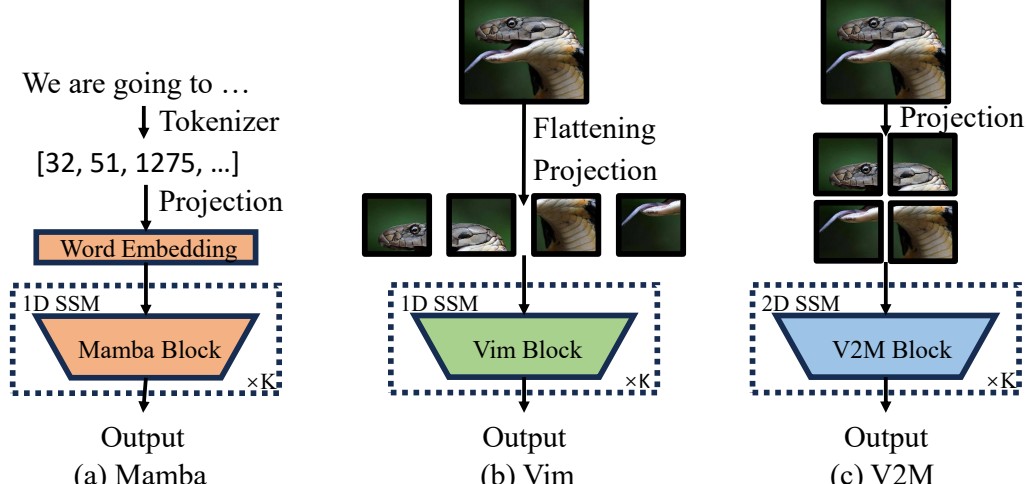

Figure 2: Comparisons between Mamba, Vim, and V2M: (a) Mamba encodes the text information after tokenization with blocks based on 1-dimensional state space models. (b) Vision Mamba (Vim) employs a straightforward flattening of the input image and then processes it through 1-dimensional state space model blocks similar to Mamba. (c) Our proposed V2M framework involves 2-dimensional state space models into the encoding blocks without the flattening operations, which is more appropriate for processing images.

that is intricately linked to the input data $\mathbf{x} \in \mathbb{R}^{B \times L \times D}$:

$$\mathbf{B} = s_B(\mathbf{x}), \quad \mathbf{C} = s_C(\mathbf{x}), \quad \mathbf{\Delta} = \tau_\Delta(s_\Delta(\mathbf{x}) + \mathbf{p}), \tag{1}$$

where $s_{\{B,C,\Delta\}}(\cdot)$ denotes the linear projection of the input data, $\tau_\Delta(\cdot)$ is the activation function, and $\mathbf{p}$ represents the learnable parameter. Subsequent discretization follows the zero-order hold (ZOH) technique.

However, the input-specific parameters render Mamba incapable of directly transitioning into a convolutional format, thereby precluding parallelization of the training process. Consequently, Mamba mitigates the decline in training velocity through parallel scanning tailored to hardware capabilities.

Mamba and 1D state space models are more congruent with the modeling of 1D input data. Conversely, for image samples, the extension of the state space model into a 2D format constitutes a more intrinsic method for feature extraction. A classical high-dimensional state space model is Roesser's state space model (Kung et al., 1977), which employs multiple hidden states to describe the high-dimensional features. The 2D form is illustrated as follows:

$$\begin{bmatrix} h^1_{i,j+1} \\ h^2_{i+1,j} \end{bmatrix} = \begin{bmatrix} \mathbf{A}_1 \mathbf{A}_2 \\ \mathbf{A}_3 \mathbf{A}_4 \end{bmatrix} \begin{bmatrix} h^1_{i,j} \\ h^2_{i,j} \end{bmatrix} + \begin{bmatrix} \mathbf{B}_1 \\ \mathbf{B}_2 \end{bmatrix} x_{i,j}, \quad y_{i,j} = [\mathbf{C}_1, \mathbf{C}_2] \begin{bmatrix} h^1_{i,j} \\ h^2_{i,j} \end{bmatrix}, \quad h_{i,j} = \begin{bmatrix} h^1_{i,j} \\ h^2_{i,j} \end{bmatrix}, \tag{2}$$

where the hidden state $h_{i,j} \in \mathbb{R}^{2N}$ is composed of two sub-states from each axis (horizontal component $h^1_{i,j}$ and vertical component $h^2_{i,j}$), the dimensions of the evolution parameters $\mathbf{A}_i$ and the projection parameters $(\mathbf{B}_i, \mathbf{C}_i)$ as well as the discretization process are equal to those in the 1D SSM. In this paper, we employ an input-contextualized discrete 2D SSM to establish the foundational architecture for image backbones.

### 3.2 2-DIMENSIONAL SSM

The form of the aforementioned 2D state space model is time-invariant, i.e., the evolution parameters and projection parameters are independent of the input. However, the transmutation of the 2D SSM into the input-dependent form precludes the direct application of convolutional training methodologies, and concurrently presents difficulties in leveraging hardware acceleration through parallel scanning processes. Accordingly, we endeavor to convert the time-varying 2D SSM into an

equivalent or near-equivalent 1D SSM, subsequently redeploying the parallel scanning methodology of 1D Mamba for the purpose of expediting the training process.

The state transition equation in Eq. (2) can be expanded as follows:

$$h^1_{i,j+1} = \mathbf{A}_1 h^1_{i,j} + \mathbf{A}_2 h^2_{i,j} + \mathbf{B}_1 x_{i,j}, \tag{3}$$

$$h^2_{i+1,j} = \mathbf{A}_3 h^1_{i,j} + \mathbf{A}_4 h^2_{i,j} + \mathbf{B}_2 x_{i,j}. \tag{4}$$

We expound upon the state transition of the horizontal component as an illustrative example, which can be decomposed into $h^1_{i,j+1} = (h^1_{i,j+1})' + (h^1_{i,j+1})''$, detailed as follows:

$$(h^1_{i,j+1})' = \mathbf{A}_1 h^1_{i,j} + \mathbf{B}_1 x_{i,j}, \tag{5}$$

$$(h^1_{i,j+1})'' = \mathbf{A}_2 h^2_{i,j}. \tag{6}$$

We observe that Eq. (5) concurs with the format of the 1D SSM, thus we may seamlessly redeploy the encoding blocks from the 1D Mamba for computation with the integration of parallel scanning. The trainable parameters are $\mathbf{A}_1$ and $\mathbf{B}_1$, respectively. Specifically, for the input samples $\mathbf{x} \in \mathbb{R}^{B \times H \times W \times D}$, the computational procedure is conducted sequentially by rows. Therefore, we process each row of features from the input sample as an independent sample for SSM computation, that is, fusing the $B$ and $H$ dimensions of the input sample, denoted as $\mathbf{x}' \in \mathbb{R}^{(BH) \times W \times D}$.

Under such circumstances, we utilize the state component values computed via Eq. (5) as new input values $x'_{i,j}$ at the corresponding positions and subsequently recombine them with Eq. (6):

$$h^1_{i,j+1} = \mathbf{A}_2 h^2_{i,j} + \mathbf{I} x'_{i,j}. \tag{7}$$

In Eq. (7), the precise computation of $h^2_{i,j}$ is intertwined with both state components simultaneously. We thus employ a judicious simplification, restricting the computation of $h^2_{i,j}$ to engage solely with the vertical aspect of the state, illustrated as follows:

$$h^2_{i+1,j} = \mathbf{A}_2 h^2_{i,j} + \mathbf{I} x'_{i,j}. \tag{8}$$

**Analysis of the simplification.** In terms of computational accuracy, we admit that calculating $h^2_{i,j}$ with only the vertical component is less precise than using the full component as specified. However, in Eq. 8, the iterative calculation process of $h^2_{i,j}$ also includes the input term $x'_{i,j}$, which is derived from Eq. 5 and inherently incorporates modeling information from the horizontal direction. This term also represents a distinct difference between the horizontal and vertical components. Therefore, the calculation of $h^2_{i,j}$ only reduces the weight of the horizontal input to some extent without completely losing all horizontal information, and the network may also strengthen the learning of the horizontally low-weighted components through the optimization of $\mathbf{A}_2$. Additionally, the direct consequence of this simplification is that it enables the use of hardware optimization algorithms for 1D SSM to achieve acceleration, which facilitates the training process. In the future, we will also explore methods for performing precise 2D SSM calculations on hardware.

Evidently, Eq. 8 can also correspond to the 1D SSM, requiring only that each column of the input sample be treated as an individual feature, with $B$ and $W$ dimensions integrated for subsequent computation, denoted as $\mathbf{x}' \in \mathbb{R}^{(BW) \times H \times D}$. Note that the projection parameter is set to $\mathbf{I}$, and the trainable parameters are confined to $\mathbf{A}_2$. Then $h^1_{i,j+1}$ can be directly derived from $h^2_{i,j}$ without additional computational processes.

**Remark.** The aforementioned simplified computational process may result in a less precise state calculation outcome, yet the state variables at each position remain coherent with the 2D SSM, encompassing both the horizontal and vertical state components simultaneously. Furthermore, the computation of each state component (i.e., the state transition equation) is intricately intertwined with the two preceding state components. Therefore, in practical applications, we are only required to separately endow the row and column features of the input sample with distinct sets of learnable parameters ( $(\mathbf{A}_1, \mathbf{B}_1)$ and $(\mathbf{A}_2, \mathbf{I})$, respectively) and subsequently conduct individual 1D SSM processing on them. Additionally, the specific implementation of Eq. (4) is analogous, differing only in the order of SSM computation for the sample columns and then the sample rows, as well as the differing learnable parameters. Specially, we present a comparison diagram in Figure 2 to highlight the differences with previous backbones.

Figure 3: An overall framework of the proposed V2M approach. We employ the 2D SSM starting from four directions (upper left, upper right, lower right, and lower left) to conduct feature encoding. Within each V2M block, we initiate the process by calculating the 2D-SSM, subsequently leveraging a MLP for feature mapping. The output features from the final V2M block are utilized for classification or downstream heads

### 3.3 VISUAL 2D MAMBA

The aforementioned implementation of the 2D SSM constitutes our Visual 2D Mamba (V2M) block. We provide an overall framework of our V2M, illustrated in Figure 3. Given an input batch $\mathbf{x}' \in \mathbb{R}^{B \times h \times w \times c}$, we first utilize the patch embedder to transform it into two-dimensional image patches and project to latent dimension, denoted as $\mathbf{x} \in \mathbb{R}^{B \times H \times W \times D}$, where $H/h = W/w$ represents the patch size. Considering the non-temporal nature of image data, we commence the 2D SSM computation from the four corners of the image as the starting positions. Concerning the implementation, we perform four-directional rotations of the embedding of the sample $\mathbf{x}$ around the central point, with respective angles of $0$, $\pi/2$, $\pi$, and $3\pi/2$:

$$\mathbf{x}_i = \mathrm{rot}_{(\pi/2) \cdot \mathrm{i}}(\mathbf{x}), \quad i = 0, 1, 2, 3, \tag{9}$$

where $\mathbf{x}_i$ denotes the $i$th rotted embedding and $rot_\theta(\cdot)$ denotes rotating the sample by an angle of $\theta$.

Subsequently, we concatenate the rotated embeddings in accordance with the batch dimension as the input to the V2D blocks:

$$\mathbf{z} = \mathrm{concat}([\mathbf{x}_0, \mathbf{x}_1, \mathbf{x}_2, \mathbf{x}_3], \ \dim = 0). \tag{10}$$

The concatenated embeddings will undergo a comprehensive feature extraction process through K V2M blocks. Within each V2M block, the features that have been modeled by the SSM will be aggregated in alignment with their original positions prior to rotation, presented as follows: (Note that we ignore the non-linear activation and the skip connection for brevity.)

$$\mathbf{z}^k = \mathrm{SSM}_{2\mathrm{D}}(\mathbf{z}^k), \ \ \mathbf{z}^k = [\mathbf{z}_i^k], \ \ \mathbf{z}^{k+1} = \mathrm{Linear}(\mathrm{sum}[\mathrm{rot}_{2\pi - (\pi/2) \cdot \mathrm{i}}(\mathbf{z}_i^k)]), \quad i = 0, 1, 2, 3 \tag{11}$$

where $\mathbf{z}^{k+1}$ will undergo a similar rotation operation as Eq. (9) before being passed to the subsequent V2M block. We proceed to extract the class token from the output of the final V2M block, which will be passed through a classifier for subsequent supervision.

**Efficiency Analysis.** We concurrently implement hardware parallel scanning algorithms akin to those adopted in Mamba, alongside expedited computations facilitated by high-speed SRAM, and a recomputation method for backpropagation, all of which are orchestrated to uphold the efficiency of the model in computation and memory storage. In addition, despite the four disparate orientations of 2D SSM in the proposed V2M framework, the concatenation along the batch dimension preserves the parallelistic characteristics of the computational process, which fully leverages the computational advantages of the GPU to forestall severe deterioration in processing speed.

## 4 EXPERIMENTS

In this section, we conducted a series of comprehensive experiments to assess the efficacy of our V2M framework. The network was initially pretrained using V2M on the ImageNet-1K dataset (Russakovsky et al., 2015), after which its performance was assessed across various downstream tasks. Moreover, we provided in-depth ablation studies to delve into the intricacies of the effectiveness of V2M.

Table 1: Classification results of different visual backbones on ImageNet. († denotes our reproduced performances under the default settings. * denotes using the pyramid architecture for V2M.)

| Method | Backbone | Image Size | Params (M). | FLOPs (G) | Top-1 Acc (%) |
|---|---|---|---|---|---|
| ResNet-18 (He et al., 2016) | ConvNet | $224^2$ | 12 | - | 69.8 |
| DeiT-T (Touvron et al., 2021) | Transformer | $224^2$ | 6 | 1.3 | 72.2 |
| PlainMamba-L1 (Yang et al., 2024) | SSM | $224^2$ | 7 | 3.0 | 77.9 |
| EffVMamba-T (Pei et al., 2024) | SSM | $224^2$ | 6 | 0.8 | 76.5 |
| **EffV2M-T (ours)** | SSM | $224^2$ | 6 | 1.0 | **76.9** |
| EffVMamba-S (Pei et al., 2024) | SSM | $224^2$ | 11 | 1.3 | 78.7 |
| Vim-T† (Zhu et al., 2024) | SSM | $224^2$ | 7 | 1.5 | 75.8 |
| **V2M-T (ours)** | SSM | $224^2$ | 7 | 1.9 | **76.2** |
| LocalVim-T (Huang et al., 2024) | SSM | $224^2$ | 8 | 1.5 | 76.2 |
| **V2M-T + local window (ours)** | SSM | $224^2$ | 8 | 1.8 | **76.4** |
| ResNet-50 (He et al., 2016) | ConvNet | $224^2$ | 25 | - | 77.2 |
| RegNetY-4G (Radosavovic et al., 2020) | ConvNet | $224^2$ | 21 | 4.0 | 80.0 |
| DeiT-S (Touvron et al., 2021) | Transformer | $224^2$ | 22 | 4.6 | 79.9 |
| Swin-T (Liu et al., 2021) | Transformer | $224^2$ | 29 | 4.5 | 81.2 |
| PlainMamba-L2 (Yang et al., 2024) | SSM | $224^2$ | 25 | 8.1 | 81.6 |
| EffVMamba-B (Pei et al., 2024) | SSM | $224^2$ | 33 | 4.0 | 81.8 |
| Vim-S† (Zhu et al., 2024) | SSM | $224^2$ | 26 | 5.1 | 80.3 |
| **V2M-S (ours)** | SSM | $224^2$ | 26 | 5.9 | **80.5** |
| LocalVim-S (Huang et al., 2024) | SSM | $224^2$ | 28 | 4.8 | 81.2 |
| **V2M-S + local window (ours)** | SSM | $224^2$ | 28 | 5.4 | **81.3** |
| VMamba-T (Liu et al., 2024b) | SSM | $224^2$ | 30 | 4.9 | 82.6 |
| **V2M-S* (ours)** | SSM | $224^2$ | 30 | 5.4 | **82.9** |
| ResNet-101 (He et al., 2016) | ConvNet | $224^2$ | 45 | - | 78.3 |
| ResNet-152 (He et al., 2016) | ConvNet | $224^2$ | 60 | - | 78.6 |
| RegNetY-8G (Radosavovic et al., 2020) | ConvNet | $224^2$ | 39 | 8.0 | 81.7 |
| RegNetY-16G (Radosavovic et al., 2020) | ConvNet | $224^2$ | 84 | 16.0 | 82.9 |
| ViT-B/16 (Dosovitskiy et al., 2020) | Transformer | $384^2$ | 86 | 55.4 | 77.9 |
| DeiT-B (Touvron et al., 2021) | Transformer | $224^2$ | 86 | 17.5 | 81.8 |
| Swin-S (Liu et al., 2021) | Transformer | $224^2$ | 50 | 8.7 | 83.2 |
| Swin-B (Liu et al., 2021) | Transformer | $224^2$ | 88 | 15.4 | 83.5 |
| PlainMamba-L3 (Yang et al., 2024) | SSM | $224^2$ | 50 | 14.4 | 82.3 |
| VMamba-S (Liu et al., 2024b) | SSM | $224^2$ | 50 | 8.7 | 83.6 |
| **V2M-B* (ours)** | SSM | $224^2$ | 50 | 9.6 | **83.8** |

## 4.1 EXPERIMENTAL SETUP

**Datasets.** We pretrain our V2M model on the training dataset of ImageNet-1K (Russakovsky et al., 2015), which comprises 1,280,000 samples across 1,000 distinct categories. The classification performance is verified on the validation set, which consists of 50,000 images. Furthermore, we leverage the COCO (Lin et al., 2014) dataset for object detection and instance segmentation, encompassing 118K images for training, 5K for validation, and 40K for testing. The semantic segmentation task is conducted on ADE20K (Zhou et al., 2019), which comprises 150 detailed semantic categories, offering 20K, 2K, and 3K images for training, validation, and testing, respectively.

**Implementation Details.** We adopted Vision Mamba (Vim) (Zhu et al., 2024), LocalMamba (Huang et al., 2024), and VMamba (Liu et al., 2024b) as our baselines for experimental comparisons, thus maintaining consistency in our experimental configuration. We provided multiple V2M models according to their parameters, including V2M-T (Tiny), V2M-S (Small), and V2M-B (Base). Additionally, the network architecture encompassed both the plain and pyramid configurations, with

Table 2: Object detection and instance segmentation results on COCO. (* denotes using the pyramid architecture for V2M.)

| Method | Detector | Params (M). | $AP^b$ | $AP^b_{50}$ | $AP^b_{75}$ | $AP^m$ | $AP^m_{50}$ | $AP^m_{75}$ |
|---|---|---|---|---|---|---|---|---|
| ResNet-50 (He et al., 2016) | MaskRCNN@1x | 44 | 38.2 | 58.8 | 41.4 | 34.7 | 55.7 | 37.2 |
| ResNet-101 (He et al., 2016) | MaskRCNN@1x | 63 | 38.2 | 58.8 | 41.4 | 34.7 | 55.7 | 37.2 |
| ConvNeXt-T (Liu et al., 2022) | MaskRCNN@1x | 48 | 44.2 | 66.6 | 48.3 | 40.1 | 63.3 | 42.8 |
| ConvNeXt-S (Liu et al., 2022) | MaskRCNN@1x | 70 | 45.4 | 67.9 | 50.0 | 41.8 | 65.2 | 45.1 |
| ConvNeXt-T (Liu et al., 2022) | MaskRCNN@3x | 48 | 46.2 | 67.9 | 50.8 | 41.7 | 65.0 | 44.9 |
| ConvNeXt-S (Liu et al., 2022) | MaskRCNN@3x | 70 | 47.9 | 70.0 | 52.7 | 42.9 | 66.9 | 46.2 |
| Swin-T (Liu et al., 2021) | MaskRCNN@1x | 48 | 42.7 | 65.2 | 46.8 | 39.3 | 62.2 | 42.2 |
| Swin-S (Liu et al., 2021) | MaskRCNN@1x | 69 | 44.8 | 66.6 | 48.9 | 40.9 | 63.2 | 44.2 |
| Swin-T (Liu et al., 2021) | MaskRCNN@3x | 48 | 46.0 | 68.1 | 50.3 | 41.6 | 65.1 | 44.9 |
| Swin-S (Liu et al., 2021) | MaskRCNN@3x | 69 | 48.2 | 69.8 | 52.8 | 43.2 | 67.0 | 46.1 |
| VMamba-T (Liu et al., 2024b) | MaskRCNN@1x | 50 | 47.3 | 69.3 | 52.0 | 42.7 | 66.4 | 45.9 |
| **V2M-S* (ours)** | MaskRCNN@1x | 50 | **47.6** | **69.4** | **52.2** | **42.9** | **66.5** | **46.3** |
| VMamba-S (Liu et al., 2024b) | MaskRCNN@1x | 70 | 48.7 | 70.0 | 53.4 | 43.7 | 67.3 | 47.0 |
| **V2M-B* (ours)** | MaskRCNN@1x | 70 | **48.9** | **70.2** | **53.6** | **43.8** | **67.5** | **47.1** |
| VMamba-T (Liu et al., 2024b) | MaskRCNN@3x | 50 | 48.8 | 70.4 | 53.5 | 43.7 | 67.4 | 47.0 |
| **V2M-S* (ours)** | MaskRCNN@3x | 50 | **49.0** | **70.6** | **53.5** | **43.8** | **67.5** | **47.2** |
| VMamba-S (Liu et al., 2024b) | MaskRCNN@3x | 70 | 49.9 | 70.9 | 54.7 | 44.2 | 68.2 | 47.7 |
| **V2M-B* (ours)** | MaskRCNN@3x | 70 | **50.0** | **70.9** | **54.8** | **44.3** | **68.4** | **47.8** |

the latter denoted by *. For image classification, we incorporated a suite of data augmentation techniques, including random cropping, random horizontal flipping, label smoothing regularization, mixup (Zhang et al., 2018), and random erasing. We adopted AdamW as the optimizer with a momentum of 0.9 and a weight decay of 0.05. We set the batch size to 512 and the training epoch to 300. We set the learning rate to $5e^{-4}$ with a cosine schedule. The input images of both training and validation were cropped to 224×224. We reported both top-1 accuracy and FLOPs for evaluation.

As for object detection and instance segmentation, we adopted the MMDetection (Chen et al., 2019) codebase for evaluation. We employed Mask R-CNN detector (He et al., 2017) with both 1x and 3x training schedules, similar to VMamba. We reported AP with different settings for comparison. Additionally, we equipped the pretrained models with UPerNet (Xiao et al., 2018) for semantic segmentation using the MMSegmentaion (Contributors, 2020) codebase. We applied AdamW with a learning rate of $6e^{-5}$ and a weight decay of 0.01. We adopted a learning schedule of 160k reported mIoU for comparison. All our experiments were performed on 8 RTX 3090 GPUs.

## 4.2 MAIN RESULTS

**Image Classification on ImageNet.** We provided the evaluation results of V2M on ImageNet classification in Table 1. The comparative methods include convolutional networks (ConvNets), transformer-based models, and the SSM baselines. We observe that SSM-based methods exhibit comparative superiority over ConvNets and transformer-based models under comparable parameters. For example, our proposed V2M-T achieved a 6.4% increase in Top-1 accuracy compared to ResNet-18 and a 4.0% improvement over DeiT-T. Furthermore, V2M-S* outperforms ResNet-50, RegNetY-4G, and DeiT-S, with respective increases of 5.7%, 2.9%, and 3.0% in Top-1 accuracy. In addition to ConvNets and transformer-based models, V2M also demonstrates a certain performance improvement across both the tiny and small model parameter configurations in comparison to the adopted baselines, registering a 0.4%/0.2%/0.3% enhancement in the tiny model and a 0.2%/0.1%/0.2% improvement in the small model compared with Vim/LocalMamba/VMamba, respectively. The experimental results verify the effectiveness of the proposed V2M framework, which indicates that the adaptive modeling process of image data through 2D SSM is more suitable compared to the straightforward flattening of images with the 1D SSM modeling.

Table 3: Semantic segmentation results on ADE20K. (* denotes using the pyramid architecture.)

| Method | Segmentor | Image size | Params (M). | mIoU (SS) | mIoU (MS) |
|---|---|---|---|---|---|
| Swin-T (Liu et al., 2021) | UperNet@160k | $512^2$ | 60 | 44.4 | 45.8 |
| Swin-S (Liu et al., 2021) | UperNet@160k | $512^2$ | 81 | 47.6 | 49.5 |
| Vim-T (Zhu et al., 2024) | UperNet@160k | $512^2$ | 13 | 41.0 | - |
| **V2m-T (ours)** | UperNet@160k | $512^2$ | 13 | **41.4** | **42.0** |
| Vim-S (Zhu et al., 2024) | UperNet@160k | $512^2$ | 46 | 44.9 | - |
| **V2M-S (ours)** | UperNet@160k | $512^2$ | 46 | **45.1** | **46.1** |
| LocalVim-T (Huang et al., 2024) | UperNet@160k | $512^2$ | 36 | 43.4 | 44.4 |
| **V2M-T + local window (ours)** | UperNet@160k | $512^2$ | 36 | **43.5** | **44.6** |
| LocalVim-S (Huang et al., 2024) | UperNet@160k | $512^2$ | 58 | 46.4 | 47.5 |
| **V2M-S + local window (ours)** | UperNet@160k | $512^2$ | 58 | **46.6** | **47.7** |
| VMamba-T (Liu et al., 2024b) | UperNet@160k | $512^2$ | 62 | 47.9 | 48.8 |
| **V2M-S* (ours)** | UperNet@160k | $512^2$ | 62 | **48.2** | **49.0** |
| VMamba-S (Liu et al., 2024b) | UperNet@160k | $512^2$ | 82 | 50.6 | 51.2 |
| **V2M-B* (ours)** | UperNet@160k | $512^2$ | 82 | **50.8** | **51.3** |

Table 4: Performances on long sequences. (LS denotes finetuning with long sequences.)

| Method | Image Size | Params (M). | Top-1 Acc |
|---|---|---|---|
| Vim-T | $224^2$ | 7 | 75.8 |
| Vim-T (LS) | $224^2$ | 7 | 78.3 |
| V2M-T | $224^2$ | 7 | 76.2 |
| V2M-T (LS) | $224^2$ | 7 | 78.8 |
| Vim-S | $224^2$ | 26 | 80.3 |
| Vim-S (LS) | $224^2$ | 26 | 81.4 |
| V2M-S | $224^2$ | 26 | 80.5 |
| V2M-S (LS) | $224^2$ | 26 | 82.0 |

Table 5: Computation speed of V2M. (TP denotes the throughput.)

| Method | Param (M). | TP. (img/s) | Top-1 Acc |
|---|---|---|---|
| Vim-T | 7 | 1624 | 75.8 |
| V2M-T | 7 | 1311 | 76.2 |
| Vim-S | 26 | 733 | 80.3 |
| V2M-S | 26 | 551 | 80.5 |
| VMamba-T | 30 | 1508 | 82.6 |
| V2M-S* | 30 | 1189 | 82.9 |
| VMamba-S | 50 | 796 | 83.6 |
| V2M-B* | 50 | 602 | 83.8 |

**Transfer Learning on Object Detection and Instance Segmentation.** We evaluated the transferability of V2M on the COCO dataset, including object detection and instance segmentation tasks. The experimental results are presented in Table 2. Consistent with the image classification on ImageNet, the SSM-based models demonstrate superior performances on object detection and instance segmentation tasks compared to transformer-based counterparts with equivalent model parameters. Moreover, compared with the corresponding baselines, the proposed V2M method exhibits certain advantages in both object detection and instance segmentation. Specifically, V2M-S* outperforms VMamba by 0.3 box AP and 0.2 mask AP under 1x schedule, which demonstrates the superior generalization ability of the representations learned by V2M.

Table 6: Effect of 2D SSM directions with V2M-T and Vim-T. (UL=Upper Left, UR=Upper Right, LL=Lower Left, LR=Lower Right)

| Direction | V2M-T Acc | Vim-T Acc |
|---|---|---|
| UL | 75.2 | 75.1 |
| UL + LR | 75.9 | 75.8 |
| UL + LR + UR | 75.9 | - |
| UL + LR + UR + LF | **76.2** | 75.9 |

**Transfer Learning on Semantic Segmentation.** We provided the semantic segmentation performances on ADE20K in Table 3. The comparative backbones include ResNet-50, ResNet-101, ConvNeXts, and Swin Transformers. We observe that V2M achieves consistent improvements over these backbones under approximate parameters. Furthermore, V2M-S* surpasses the VMamba-T baseline by 0.3 mIoU (SS), demonstrating the effectiveness of the proposed framework.

### 4.3 EXPERIMENTAL ANALYSIS

**Effect of Modeling Directions of the 2D SSM.** Our proposed V2M method undergoes 2D SSM modeling in four directions, and thus we assess the impact of different directions on model per-

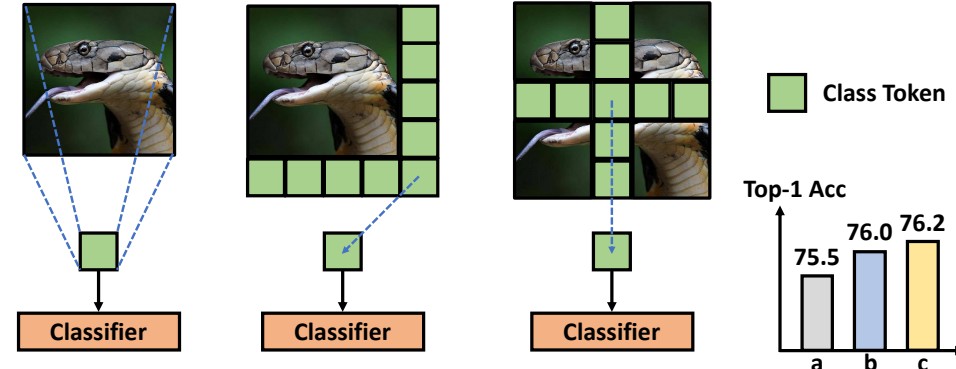

Figure 4: Arrangements of the class token. (a) Obtaining a class token through feature mean pooling. (b) Adding an additional row and column of class tokens at the edge of the image and adopting the corner token for subsequent classification. (c) Adding an additional row and column of class tokens at the middle of the image and adopting the center token for subsequent classification.

formance. Specifically, we individually attempted modeling in 1, 2, 3, and 4 different directions, and subsequently tested the classification performance on ImageNet. The comparison results are illustrated in Table 6. We discern that the performance of V2M on the classification task ascends incrementally with the increase of modeling directions, culminating in optimal outcomes upon the integration of modeling across all four directions.

**Performance on Long Sequences.** We tested the performance of V2M on long sequences by referring to the settings of Vim (Zhu et al., 2024), presented in Table 4. Specifically, our V2M-T model achieves an accuracy of 78.8% compared to 78.3% by Vim-T in the long sequence setting for ImageNet-1K, while V2M-S further reached an accuracy of 82.0% as opposed to 81.4% by Vim-S. This demonstrates the effectiveness of the V2M framework on long sequence data.

**Computation Speed.** We present the computation speed of V2M compared with the baselines in Table 5. We admit that V2M may slow down the model to a certain extent, which is a limitation of V2M and corresponds to the increase in FLOPs as we have illustrated in Table 1. However, V2M also enhances the classification performance on the ImageNet-1K dataset. We will further optimize the speed of V2M in the future.

**Arrangements of Classification.** Diverging from the Vim baseline, our proposed V2M performs 2D SSM modeling by independently correlating the features of rows and columns without the flattening of the entire image patch. Consequently, we are unable to merely append a solitary class token for subsequent classification tasks. To address this limitation, we present three viable solutions, including obtaining the class token through feature mean pooling, augmenting the image with an additional row and column of class tokens at the edge, and incorporating a row and column of class tokens at the middle of the image, shown in Figure 4. We observe that the third scheme yields a commendable performance and is thus designated as the default setting.

## 5 CONCLUSION

In this paper, we have presented a visual 2-dimensional mamba (V2M) framework for effective image representation learning. We have employed the 2D SSM for the modeling process of images, thereby preserving the inherent prior of local invariance within the images. We have concurrently performed 2D SSM modeling in four directions to enhance the representational capacity of the model considering the non-temporal nature of the input images. We have conducted a series of experiments that encompass classification and various downstream tasks, including object detection, instance segmentation, and semantic segmentation, to assess the effectiveness of our framework. Experimental results have demonstrated the superior classification as well as transferability performances of the proposed V2M framework.

**Limitations.** The implementation of 2D SSM modeling in four directions within each V2M block necessitates a compromise in speed. Future research will concentrate on both software and hardware algorithm optimizations to mitigate this drawback.

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
