# OpenReview forum: "V2M: Visual 2-Dimensional Mamba for Image Representation Learning"
_ICLR.cc/2025/Conference — Submitted to ICLR 2025_

### Official Review · Reviewer_5M99 · 2024-10-31

**Soundness:** 3
**Presentation:** 3
**Contribution:** 3
**Rating:** 5
**Confidence:** 5

**Summary:**

The submission introduces a Visual 2-Dimensional Mamba (V2M) model, aimed at preserving the inherent prior of 2D image data. The V2M model incorporates a 2D State Space Model (SSM) that directly processes image tokens in 2D state space by considering two adjacent states in both dimensions. The authors convert the time-varying 2D SSM into a near-equivalent 1D Mamba format for efficient parallel computation. Experimental results show that V2M outperforms existing visual backbones on ImageNet classification and various downstream tasks.

**Strengths:**

1. The proposed time-varying 2D SSM is based on Roesser’s state space model, which intuitively depicts the spatial information of the image.

2. This paper proposes a simplified computational process for Roesser’s state space model, resulting in two consecutive 1D SSM block, thus achieving hardware-efficient parallel processing.

3. The proposed V2M model adopts four scanning directions and compares three types of classification tokens: mean pooling, edge class token and middle class token.

**Weaknesses:**

1. Although the theoretical analysis of 2D SSM is based on Roesser’s state space model, it is simplified into two 1D SSM processes for parallelism. I am not sure whether this simplification retains the ability of Roesser’s SSM to capture the spatial dependencies within the image.

2. The performance improvement is tiny especially in the downstream tasks. It is not sure whether the improvement is brought by the proposed 2D SSM. From the ablation study in Table 4, modeling in four directions seems to offer greater benefits than the 2D SSM itself.

3. The FLOP of V2M is much higher than that of baseline models, for example, 1.5G for Vim-T and 1.9G for V2M-T. Specific operating speeds, such as throughput, should be provided. Please gradually add each module to the baseline model, and give the corresponding number of parameters, FLOPs, throughput and performance.

**Questions:**

1. The method part (Sec. 3.2) is not clear enough. From equation 11 and 12, it seems that the horizontal component and the vertical component are the same after simplification, which contradicts formula 6. It is better if the authors can give a clearer and comprehensive explanation about the simplified computational process.
2. Please add the ablation study for 2D SSM block itself.

---

> ### Author Response · Authors · 2024-11-23
>
> ## W1 and Q1: Analysis of the simplification
> Thanks for this constructive suggestion. In terms of computational accuracy, we admit that calculating $h^2_{i,j}$ with only the vertical component is less precise than using the full component as specified. However, the iterative calculation process of $h^2_{i,j}$ in Eq. 8 in the revised manuscript also includes the input term $x_{i,j}$, which is derived from Eq. 5 in the revised version and inherently incorporates modeling information from the horizontal direction.  This term also represents a distinct difference between the horizontal and vertical components. Therefore, the calculation of $h^2_{i,j}$ only reduces the weight of the horizontal input to some extent without completely losing all horizontal information, and the network may also strengthen the learning of the horizontally low-weighted components through the optimization of $\mathbf{A}_2$. Additionally, the direct consequence of this simplification is that it enables the use of hardware optimization algorithms for 1D SSM to achieve acceleration, which facilitates the training process. In the future, we will also explore methods for performing precise 2D SSM calculations on hardware. (Please see Section 3.2 in the revised manuscript marked in red.)
>
> ## W2: Performances
>
> Thanks for this comment. We believe that the performance improvements on the ImageNet-1K and downstream tasks should not be considered insignificant. For instance, LocalMamba achieved a ~0.2% improvement over the baseline in downstream object detection tasks.
>
> In Table 4, we employ 2D SSM in four directions to ensure that the class token can obtain effective information from all four directions. As we place the class token in the middle position in the main experiment, reducing the modeling in one direction would result in the class token lacking information from the tokens that precede it in that direction, as shown in Figure 4. This design is closely related to our 2D modeling of the image. For a more fair ablation, we conducted experiments using the mean pooling to obtain the class token and obtained an accuracy of 75.5% as shown in Figure 4. We further evaluated the performance of only using the upper-left direction and achieved a classification accuracy of 75.3%. This shows that reducing the number of directions hardly affects the performance under this setting.
>
> ## W3 and Q2: Ablation and computation speed
>
> Thanks for this nice suggestion. Our 2D SSM block is an integrated unit and cannot be disassembled into modules (for example, removing parts of the structure within V2M would prevent obtaining the final classification results). However, we have supplemented the model speed indicator (throughput) in addition to FLOPs in the revised version. We admit that V2M may slow down the model to a certain extent, which corresponds to the increase in FLOPs as we have illustrated (refer to our limitation part). However, V2M also enhances the classification performance. We will further optimize the speed of V2M in the future. (Please see Section 4.3 and Table 5 in the revised version marked in red.)

---

> > ### Comment · Reviewer_5M99 · 2024-11-26
> >
> > Thanks for your response. However, we have some follow-up concerns:
> >
> > **W1: Simplification**
> >
> > We look forward to the authors exploring methods for implementing precise 2D SSM calculations on hardware, which may fundamentally address Mamba's challenges in spatial modeling for visual data.
> >
> > **W2: Performance**
> >
> >  My question has not been fully addressed; here are further comments.
> > 1. *Small Improvement vs. Higher Cost*: If your method increases computation (e.g., higher FLOPs and slower speed) but only shows slight performance gains, it is unclear why the model is worth adopting.
> >
> > 2. *Reproducibility Issues:* In Table 1, the reproduced Top-1 accuracy for Vim-T (75.8%) is lower than the original result (76.1%), while Vim-S matches the original (80.3%). Why is there a clear drop for Vim-T but not Vim-S? We noticed that V2M-T achieves a performance of 76.2%, surpassing your Vim-T reproduction but only slightly outperforming the original Vim-T. A clear explanation is necessary to avoid unnecessary confusion.
> >
> > 3. *Performance:* If V2M reduces its four-directional scanning to bidirectional like Vim, Table 6 shows that V2M's performance will decrease from 76.2% to 75.9%. In that case, V2M's performance might become comparable to Vim. However, the proposed V2M does significantly increase Flops (1.5 for Vim-T and 1.9 for V2M-T in Table 1) and reduces throughput (1624 for Vim-T and 1311 for V2M-T in Table 5). These issues are critical factors in my evaluation of the paper.
> >
> > If these issues can be properly addressed, I would be open to revising my evaluation.

---

> > > ### Author Response · Authors · 2024-11-27
> > >
> > > ## W1: Simplification
> > >
> > > Thanks for this suggestion. We are already attempting to implement a precise 2D SSM for image feature modeling, which involves modifying the underlying CUDA code to perform a 2D parallel scanning process. This is more complex compared to the 1D parallel scanning. However, due to time constraints, we may not be able to complete the corresponding experimental validation during the rebuttal period. We also hope that you can acknowledge our work and we will gradually refine the precision of the method.
> > >
> > > ## W2: Small Improvement vs. Higher Cost
> > >
> > > Thanks for this comment. First and foremost, the key point of this paper lies in revealing the irrationality of previous vision Mambas in the modeling process and proposing a more reasonable modeling approach. Although these previous vision Mambas may have faster training and inference speeds, the simplistic flattening of image data inevitably has logical flaws. On the contrary, we believe that V2M has a higher potential and certain inspirational value. We have essentially replaced the modeling process with a 2D process, which is naturally suitable for two-dimensional image data. Although we currently have a disadvantage in terms of speed, we are also attempting to improve this through hardware optimization as part of our future work. Therefore, V2M provides a possible solution of vision Mambas and is more likely to be a reasonable and effective Mamba modeling framework.
> > >
> > > Just as researchers took a considerable amount of time to iterate before finally presenting the relatively perfect ViT and DeiT works when applying transformers from natural language processing to computer vision, a series of exploratory works also faced performance and speed issues or challenges. However, we also believe that these works have certain value. As pioneering works, they explored possible directions and avoided incorrect solutions for the final frameworks like ViT.
> > >
> > > Meanwhile, V2M can also provide some inspiration for video work, that is, to continue to increase the dimensionality of SSM to potentially achieve V3M for application in video analysis and other fields. This is obviously more reasonable than simply flattening video data into three dimensions for feature modeling.
> > >
> > > In addition, our V2M has certain advantages over some vision Mambas works. For example, LocalVim-T can only achieve an accuracy of 75.8% without conducting a scan direction search, while our Vim-T can achieve 76.2%. Although our FLOPs increased by 0.4G, the model’s parameter count was reduced by 1M and the accuracy was improved by 0.4%. Moreover, LocalVim-T, which utilized scan direction search, achieved the same classification accuracy of 76.2% as our Vim-T, but they required an additional 100 epochs for the search process. Therefore, although they may have relatively lower FLOPs, the time it takes to achieve practical results is likely to be longer than ours.
> > >
> > > We express our gratitude again for your constructive suggestions and important comments, and we also hope you can acknowledge the exploration we have undertaken towards a more rational vision Mamba modeling scheme.

---

> > > ### Author Response · Authors · 2024-11-27
> > >
> > > ## W3: Reproducibility Issues
> > >
> > > Thanks for this comment. We have conducted a detailed examination of the previous training configurations. Due to the limited number of GPUs at our disposal, in order to make more efficient use of the limited resources to verify the effectiveness of the scheme, we reduced the original batch size of 128 to 64 per card when conducting all Vim-T and V2M-T experiments. This may have led to a potential decrease in performance. However, we ensure that the hyperparameters used for Vim-T and V2M-T are exactly the same, making the comparative experiments fair. Additionally, we will also retrain and evaluate under the original settings and make the necessary modifications in the subsequent updated version of the paper.
> > >
> > > Once again, we deeply apologize for possible confusion in our paper.
> > >
> > > ## W4: Performance
> > >
> > > Thanks for this comment. Firstly, the reason V2M needs to model in four directions is similar to Vim’s bidirectional modeling. As can be seen in Figure 4 of our paper, when we place the class token in the center, if we only model from two directions (the top left and bottom right), the class token would not capture the information from the bottom left and top right tokens, making a 0.3% decrease in accuracy reasonable. Analogous to Vim, using only the forward SSM modeling would result in the class token missing relevant information from the tokens behind it. We have supplemented Table 6 with comparative experiments on Vim-T in different modeling directions in the revised version. When Vim shifts from bidirectional to unidirectional, the classification accuracy drops from 75.8% to 75.1%. Moreover, we have also tried increasing the number of directions for Vim-T to 4, which only yielded a 0.1% increase in accuracy, but the FLOPs increased from 1.5G to 1.8G, and the throughput dropped from 1624 to 1329, indicating that expanding the modeling directions on Vim does not bring about performance gains similar to V2M. Additionally, this also proves that V2M-T can achieve 0.3% higher performance than Vim-T at a similar FLOPs and throughput, demonstrating the effectiveness of V2M. Due to time constraints, we will also supplement more performance comparisons between V2M and Vim under similar conditions, such as FLOPs, in the future.

---

> ### Author Response · Authors · 2024-12-02
> **Gentle Reminder Regarding Review of Reviewer 5M99**
>
> Dear reviewer 5M99,
>
> I hope this message finds you well. We greatly appreciate the valuable feedback and suggestions you have provided so far. As the deadline approaches, we are eager to receive your feedback on our response and revisions. If possible, we kindly request an update on the progress of the review to ensure we can address any further comments or revisions promptly.
>
> Should you require any additional information or assistance from our end to help facilitate the review process, please do not hesitate to let us know. Your insights are highly valuable to us, and we genuinely appreciate your time and effort in reviewing our paper.
>
> Thank you for your patience and cooperation. We are looking forward to hearing from you soon.
>
> Warm regards,
>
> Submission2798 Authors.

---

> > ### Comment · Reviewer_5M99 · 2024-12-02
> >
> > Thank you to the authors for their thoughtful discussion and detailed responses to my questions and concerns. I truly appreciate their effort. However, as a few of my concerns remain partially unresolved, and after considering the feedback from other reviewers, I feel it is appropriate to maintain my initial score. Thank you.

---

> > > ### Author Response · Authors · 2024-12-03
> > >
> > > Thank you for your suggestions and feedback. We kindly hope that you can specifically point out the issues that remain unresolved and any other concerns you may have about the paper, so that we can further provide targeted responses and explanations. We look forward to hearing from you and accepting your further specific suggestions.

---

### Official Review · Reviewer_7MsE · 2024-11-03

**Soundness:** 3
**Presentation:** 2
**Contribution:** 2
**Rating:** 3
**Confidence:** 3

**Summary:**

The paper proposes the Visual 2-Dimensional Mamba (V2M) framework, an innovative model for image representation learning. Unlike traditional Mamba models that process image data in a 1D sequence, V2M adapts a 2D state space model to retain spatial structure, preserving local relationships between pixels. This 2D adaptation allows V2M to process image patches as 2D entities directly, thereby enhancing locality and coherence in visual representation.

**Strengths:**

This paper innovatively introduces a 2D state space to address the spatial coherence issues faced by SSM models in the visual domain, achieving promising results.

**Weaknesses:**

1.Novelty

The novelty is moderate, as there has already been substantial research, such as VMamba[1], exploring how sequence models can preserve 2D structural information in visual tasks.

2.Implementation

While this paper introduces a theory of 2D state space to address coherence issues in visual tasks,  its implementation relies on excessive simplification, lacking a detailed explanation of the rationale behind this approach and its potential implications.

3.Performance

The performance demonstrated in this paper shows only minor gains over early baselines like Vim[2] and VMamba[1], without advantages over more recent models like EfficientVMamba[3].

4.Experiment

It is foreseeable that the proposed method will impact speed and memory usage; however, the paper lacks sufficient efficiency tests, memory usage tests, and ablation studies, resulting in limited experimental support.

5.Presentation

The expression of the formulas is somewhat confusing, such as the approach used to split Equation 7 into Equations 9 and 10, and the unclear logical progression from Equation 11 to Equation 12.

[1]VMamba: Visual State Space Model. Proceedings of the 38th Conference on Neural Information Processing Systems.

[2]Vision Mamba: Efficient Visual Representation Learning with Bidirectional State Space Model. Proceedings of the 41st International Conference on Machine Learning.

[3]EfficientVMamba: Atrous Selective Scan for Light Weight Visual Mamba. arXiv preprint arXiv:2403.09977.

**Questions:**

Could you add more efficiency experiments and ablation studies to this paper and provide a more thorough discussion based on those findings?

Additionally, could you offer a more detailed analysis and explanation of the simplifications mentioned in Section 3.2?

---

> ### Author Response · Authors · 2024-11-23
>
> ## W1: Novelty
> We want to clarify that although both V2M and VMamba claim to preserve the two-dimensional structural information of visual data, VMamba and other vision Mambas adopt similar measures by unfolding the images row by row and column by column, followed by modeling in these directions. Therefore, in essence, they still largely destroy the two-dimensional information during the modeling process, merely presenting a combination of one-dimensional modeling results. In contrast, V2M alters the modeling logic at the lower level, utilizing two-dimensional state space equations for feature extraction of image data. Although we have simplified the modeling of the two-dimensional state space equations in our implementation to obtain some one-dimensional results, we do so only to leverage the acceleration process of one-dimensional SSM and have eliminated the operation of flattening images by rows and columns (we no longer involve the operation of concatenating tokens that are spatially distant from each other). In general, V2M fundamentally represents a more suitable modeling method for two-dimensional images
> ## W2 and Q2: Simplification
> Thanks for this comment. In terms of computational accuracy, we admit that calculating $h^2_{i,j}$ with only the vertical component is less precise than using the full component as specified. However, the iterative calculation process of $h^2_{i,j}$ in Eq. 8 in the revised manuscript also includes the input term $x_{i,j}$, which is derived from Eq. 5 in the revised version and inherently incorporates modeling information from the horizontal direction. This term also represents a distinct difference between the horizontal and vertical components. Therefore, the calculation of $h^2_{i,j}$ only reduces the weight of the horizontal input to some extent without completely losing all horizontal information, and the network may also strengthen the learning of the horizontally low-weighted components through the optimization of $\mathbf{A}_2$. Additionally, the direct consequence of this simplification is that it enables the use of hardware optimization algorithms for 1D SSM to achieve acceleration, which facilitates the training process. We will also explore methods for performing precise 2D SSM calculations on hardware. (Section 3.2 in the revised manuscript marked in red.)
> ## W3: Performances
> We believe that an improvement of 0.2%-0.4% in classification results on the ImageNet-1K dataset should not be considered marginal. Many existing works (e.g., LocalMamba) have also achieved similar improvements over the baseline as our method. Additionally, our method is plug-and-play and compatible with existing Vision Mamba backbones. We also conducted comparative experiments by incorporating our method into EfficientVMamba. We achieved a 0.4% improvement (to 76.9%) over EffVMamba-T (76.5%) on the ImageNet-1K dataset, demonstrating the applicability of our method. (Please see Table 1 in the revised manuscript.)
> ## W4 and Q1: Computation speed
> Thanks for this nice suggestion. In terms of memory, we exercised strict control during the model design phase, resulting in a parameter count consistent with the baseline and thus no significant difference in memory usage. As for the speed-related metrics of the model, we have provided the corresponding FLOPs in our paper, which to some extent reflects the complexity of our algorithm. In addition, we have conducted speed assessments for V2M under both Vim and VMamba backbones, with throughput as the metric. We admit that V2M may slow down the model to a certain extent, which corresponds to the increase in FLOPs as we have illustrated (refer to our limitation part). However, V2M also enhances the classification performance. We will further optimize the speed of V2M in the future. (Please see Section 4.3 and Table 5 in the revised version marked in red.)
> ## W5: Presentation
> We apologize for possible misunderstanding. Splitting Equation 7 (Eq. 3 in the revised manuscript) into Equations 9 (Eq. 5 in the revised manuscript) and 10 (Eq. 6 in the revised manuscript) merely consists of calculating the first and third terms, as well as the second term of Equation 7 separately. That is, Equation 9 corresponds to the first and third terms of Equation 7, while Equation 10 corresponds to the second term. For the transition from Equation 11 to Equation 12, we focus on the computation of $h^2_{i,j}$, which is where the simplification of our method lies—in restricting the calculation of $h^2_{i,j}$ to be related only to the vertical aspect of the state and not to $h^1_{i,j}$. Therefore, Equation 12 is the derivation of the evolution of $h^2_{i,j}$. We first obtain the result of the first term in Equation 11 through Equation 12, and then add it to the second term of Equation 11 to get the final output. We may not have described this process very clearly in our paper, and for this, we apologize again and have provided further elaboration in the revised version.

---

> > ### Comment · Reviewer_7MsE · 2024-11-26
> >
> > Regarding novelty, I generally agree with your perspective, but I believe "moderate" is an appropriate evaluation.This is because the problem has already been proposed and widely discussed, and the new method presented in the paper essentially remains within the framework of the multi-directional scanning approach.
> >
> > Regarding the update of the horizontal and vertical state components, I understand that your simplification is aimed at leveraging the original acceleration implementation of the 1D SSM as much as possible. However, the rationality of this simplification still warrants experimental analysis or detailed derivation. Relying solely on intuition for such macro-level simplifications is likely to lead to pitfalls.
> >
> > Regarding model performance, first of all, I appreciate the timely addition of experiments on EfficientVMamba. In fact, I believe the improvement presented in this paper is not particularly significant due to the following reasons:
> >
> > 1. The comparisons in the paper are made under the premise of significantly increased computational cost (e.g., V2M-T has 1.9G FLOPs, whereas Vim-T has only 1.5G FLOPs).
> >
> > 2. Table 1 in the paper does not use the results reported in the original Vim paper as a baseline but instead uses the results reproduced by your team. In fact, I have conducted experiments on Vim, and the accuracy reported in the original paper is reproducible.
> >
> > 3. Vim and VMamba were early works developed shortly after Mamba's introduction. Therefore, achieving an approximately 0.2% accuracy improvement on ImageNet-1K based on them is not particularly significant. However, your supplementary experiments on EfficientVMamba to some extent address this issue.
> >
> > Regarding computational speed, I appreciate the timely addition of throughput metric tests. I look forward to seeing you address the speed issues of V2M in the future.
> >
> > Regarding the issues with the presentation in Section 3.2, although you were able to explain the equations in your response, I believe it would be better to modify the relevant notation in the paper to make the logic clearer. For example, after splitting Equation (7) into (9) and (10), the left-hand side of the equation should not remain the same but should be replaced with new symbols. This was the original intent behind my question.
> >
> > Overall, I believe the solution proposed in this work is quite interesting and warrants further exploration.

---

> > > ### Author Response · Authors · 2024-11-27
> > >
> > > ## W1: Novelty
> > >
> > > Thanks for this comment. Although the Mamba modeling for images has already been proposed and widely discussed, V2M is completely different from all previous methods but not essentially remains within the framework of the multi-directional scanning approach.
> > >
> > > Whether it’s LocalMamba or VMamba, and a series of other methods, although they claim to be 2D modeling processes, they merely involve a simple expansion by row followed by an expansion by column, or various combinations of the two. Specifically, for an embedding of size B\*H\*W\*D, they would process it into a sequence of B\*(HW)\*D or B\*(WH)\*D and then perform 1D modeling on the (HW) or (WH) dimension. Essentially, this still results in tokens that are far apart in space being placed in adjacent positions during the modeling process (for example, the tokens at positions (1,0) and (0,W-1) are not adjacent and are likely not related, but when expanded by row, the token at position (1,0) will be placed next to the token at position (0,W-1)).
> > >
> > > In contrast, V2M does not have this issue at all. Even in our simplified modeling process, the embedding is only processed in the forms of (BH)\*W\*D and (BW)\*H\*D, followed by a 1D SSM process on the W or H dimension, and this process does not have the problem of tokens that are spatially distant being in adjacent modeling sequences. This is the core issue of the 2D image modeling in Mamba.
> > >
> > > Therefore, our method is a fundamental solution and exploration of this issue, which previous methods did not possess. Additionally, if there have been papers that utilized a solution similar to our V2M, we hope you can list them for us to conduct more targeted comparisons and explanations.
> > >
> > > Finally, we once again appreciate for your suggestions and strict requirements for our paper, and we hope our explanation can give you a clearer understanding of the core novelty of V2M, and that you can recognize our work that explores a fundamental solution to this problem.
> > >
> > > ## W2: Simplification
> > > Thanks for this suggestion. Firstly, our simplification process is indeed aimed at utilizing the relevant operations of 1D SSM for algorithm acceleration to some extent. However, we reiterate that our overall modeling process is fundamentally different from previous vision mamba papers, as illustrated before. Additionally, to experimentally demonstrate that our simplification process does not lead to the same effects as previous vision mamba baselines, we conducted simple flattening operations and 1D SSM modeling in four directions separately. This means extending the bidirectional process in Vim-T to four directions. We found that this approach achieves a classification accuracy of 75.9% on ImageNet-1K, which is 0.3% lower than the accuracy of 76.2% for V2M-T. This also indicates that our simplification process still enhances the modeling ability of the image data.
> > >
> > > ## W3: Performance -- Increased computational cost
> > >
> > > Thanks for this comment. Firstly, we would like to clarify that the relatively slower speed of V2M is primarily due to the need for a 2D modeling process in four directions, the reason of which is similar to Vim’s bidirectional modeling. As can be seen in Figure 4 of our paper, when we place the class token in the center, if we only model from two directions (the top left and bottom right), the class token would not capture the information from the bottom left and top right tokens, making a 0.3% decrease in accuracy reasonable. Analogous to Vim, using only the forward SSM modeling would result in the class token missing relevant information from the tokens behind it. We have supplemented Table 6 with comparative experiments on Vim-T in different modeling directions in the revised version. When Vim shifts from bidirectional to unidirectional, the classification accuracy drops from 75.8% to 75.1%. Moreover, we have also tried increasing the number of directions for Vim-T to 4, which only yielded a 0.1% increase in accuracy, but the FLOPs increased from 1.5G to 1.8G, and the throughput dropped from 1624 to 1329, indicating that expanding the modeling directions on Vim does not bring about performance gains similar to V2M. Additionally, this also proves that V2M-T can achieve 0.3% higher performance than Vim-T at a similar FLOPs and throughput, demonstrating the effectiveness of V2M.

---

> ### Author Response · Authors · 2024-11-27
>
> ## W4: Performance -- Reproduced results of Vim-T
>
> Thanks for this comment. We have conducted a detailed examination of the previous training configurations. Due to the limited number of GPUs at our disposal, in order to make more efficient use of the limited resources to verify the effectiveness of the scheme, we reduced the original batch size of 128 to 64 per card when conducting all Vim-T and V2M-T experiments. This may have led to a potential decrease in performance. However, we ensure that the hyperparameters used for Vim-T and V2M-T are exactly the same, making the comparative experiments fair. Additionally, we will also retrain and evaluate under the original settings and make the necessary modifications in the subsequent updated version of the paper. We deeply apologize for possible confusion in our paper.
>
> ## W5: Performance -- Improvements over baselines
>
> Thanks for this comment. V2M is a plug-and-play method that can be applied to various vision Mamba frameworks, not limited to Vim and VMamba, among others. We conducted experiments on Vim, VMamba, and LocalMamba because they are representative works in the vision Mamba series and have attracted considerable attention. Additionally, we have demonstrated that V2M can further enhance the performance of EfficientVMamba as you have mentioned and acknowledged, and we will also explore applying V2M to more baselines in the future.
>
> ## W6: Computational speed
>
> Thanks for this comment and your interest of our work. We are modifying the underlying CUDA code to perform a 2D parallel scanning process, which helps in precise calculations and accelerates the V2M framework.
>
> ## W7: Presentation
>
> Thanks for this suggestion. We have further modified our presentation of these equations in the revised manuscript.
>
> ## Summary
>
> In general, we are grateful for your valuable comments on our work and the interesting aspects you mentioned about the V2M solution. We have provided as detailed an explanation and analysis as possible for the shortcomings you pointed out, and we will continue to explore further in the future. Therefore, we hope that you can reassess our work, which will also encourage and support our subsequent explorations.

---

> ### Author Response · Authors · 2024-12-02
> **Gentle Reminder Regarding Review of Reviewer 7MsE**
>
> Dear reviewer 7MsE,
>
> I hope this message finds you well. We greatly appreciate the valuable feedback and suggestions you have provided so far. As the deadline approaches, we are eager to receive your feedback on our response and revisions. If possible, we kindly request an update on the progress of the review to ensure we can address any further comments or revisions promptly.
>
> Should you require any additional information or assistance from our end to help facilitate the review process, please do not hesitate to let us know. Your insights are highly valuable to us, and we genuinely appreciate your time and effort in reviewing our paper.
>
> Thank you for your patience and cooperation. We are looking forward to hearing from you soon.
>
> Warm regards,
>
> Submission2798 Authors.

---

### Official Review · Reviewer_pHLf · 2024-11-04

**Soundness:** 3
**Presentation:** 2
**Contribution:** 3
**Rating:** 8
**Confidence:** 5

**Summary:**

The paper presents Visual 2-Dimensional Mamba (V2M), a novel framework for image representation learning that adapts the state space model (SSM), specifically the Mamba model, to the two-dimensional structure of image data. This paper extends the 1d SSM to a 2d form to fundamentally accommodate the structure of image data. As a non-hierarchical vision mamba method, it achieves better results than its baseline Vim.

**Strengths:**

- The consideration of 2d spatial relationship in the visual domain is intuitive and sounds reasonable.
- Different from other hierarchical vision mamba methods, this paper handles the visual perception with a plain, non-hierarchical architecture, maintaining the ability in multimodality applications.
- Compared with the baseline method, this paper presents significant improvements in both classification and dense prediction tasks.
- This paper aims at a good question. Scanning strategy with SSM methods is a key problem because the 1d original scanning does not treat all tokens equally.

**Weaknesses:**

- The character corners in Figure 1 require further explanation.
- Note that function names in formulas are usually typeset properly in Roman font, e.g., `rot`, `concat`, `SSM`, `Linear`, `sum` in Eq. 13~15.
- The paper references preprints and arXiv versions of significant works, such as Mamba (COLM), Vision Mamba (ICML), and VMamba (NeurIPS). The authors should update these citations to their final published versions to reflect the current state of the literature.

**Questions:**

Please refer to the weakness part.

---

> ### Author Response · Authors · 2024-11-23
>
> ## W1: Character corners
>
> Thanks for this suggestion and we apologize for possible misunderstanding. The top right corner of the characters in Figure 1 (1 and 2) represents the dimensions of the state space equation. We have extended the one-dimensional state space equation to two dimensions, modeling it from both the horizontal and vertical aspects (1 denotes horizontal and 2 denotes vertical). The bottom right corner (i and j) represents the spatial position of the state, for example, the horizontal state at position (i, j) is determined by the horizontal state and vertical state at position (i, j-1). We have provided these explanations in the revised version. (Please see Figure 1 in the revised manuscript marked in red.)
>
> ## W2: Function names in formulas
>
> Thanks for this comment. We have adopted the Roman font for function names in formulas in the revised version. (Eq. (9-11) in the revised version.)
>
> ## W3: Citations
>
> Thanks for this nice suggestion. We have updated the citations to their final publication in our revised manuscript.

---

> ### Comment · Reviewer_pHLf · 2024-11-24
>
> The authors' rebuttal resolves my concerns. And I keep my scores.

---

### Official Review · Reviewer_RmnR · 2024-11-10

**Soundness:** 3
**Presentation:** 3
**Contribution:** 3
**Rating:** 6
**Confidence:** 4

**Summary:**

The paper proposes a 2D state-space model (SSM) for visual representation learning, named V2M. Unlike previous Mamba-based visual representation methods that perform 1D sequence learning, V2M operates in 2D space, with each hidden state conditioned on its top and left tokens. Baron et al. have presented a 2D SSM but neglect the input-dependent characteristic of SSM, which is crucial for boosting performance.

**Strengths:**

- Developing high-performance, high-dimensional state-space models is an important topic, and the paper is an interesting attempt in this direction.

- The results in the paper are generally good, demonstrating strong performance on ImageNet classification, COCO detection, and segmentation compared to strong baselines such as ViM, LocalMamba, and VMamba.

**Weaknesses:**

- One of the essential advantages of Mamba is its high efficiency in processing long sequences. The paper does not show V2M's performance on long sequences.

- V2M's runtime is not reported. Given that hardware-efficient implementation is an important part of V2M, comparing its runtime with 1D Mamba is crucial.

**Questions:**

- Writing: Use "Eq.~\eqref{xxx}" for equation references. The background formulation part can be shortened, e.g., Eqs. (1-5), as these have been widely described in previous vision Mamba papers.

- V2M seems better at spatial information modeling compared to ViM. If the positional embedding in V2M is removed, how significant is the performance drop? The performance drop could be minimal, right?

- How is the classification head designed?

- It would be interesting to see the formulation and results of V3M for video modeling.

**Details Of Ethics Concerns:**

No Ethics Concerns

---

> ### Author Response · Authors · 2024-11-23
>
> ## W1: V2M's performance on long sequences
>
> Thanks for this suggestion. We conducted long sequence fine-tuning with reference to Vision Mamba (Vim), ensuring a training epoch of 30 for comparison purposes. Specifically, our V2M-T model achieved an accuracy of 78.8% compared to 78.3% by Vim-T in the long sequence setting for ImageNet-1K, while V2M-S further reached an accuracy of 82.0% as opposed to 81.4% by Vim-S. This demonstrates the effectiveness of the V2M framework on long sequence data. Due to time constraints, we plan to explore the long sequence settings with more backbone architectures in the future. (Please see the revisions in Section 4.3 and Table 4 marked in red.)
>
> ## W2: Computation speed
>
> Thanks for this suggestion. We have conducted speed assessments for V2M during the rebuttal under both Vim and VMamba backbones with throughput as the metric as Table 5 in the revised version. We provided the corresponding FLOPs in the orginal paper, which to some extent reflects the complexity of our algorithm. We acknowledge that V2M may slow down the model to a certain extent, which corresponds to the increase in FLOPs as we have illustrated (refer to our limitation part). However, V2M also enhances the classification performance on the ImageNet-1K dataset. We will further optimize the speed of V2M in the future. (Please see the revisions in Section 4.3 and Table 5 marked in red.)
>
> ## Q1: Equation references and background formulation
>
> Thanks for this nice suggestion. We have utilized the correct equation references and shortened the background formulation part in the revised version of our paper.
>
> ## Q2: Positional embedding
>
> Thanks for this comment. We have verified the effectiveness of V2M on spatial information modeling by removing the positional embedding compared with Vim. Specifically, we have tested the corresponding performance drops of V2M-T and V2M-S without the positional embedding, which is 76.0% (-0.2%) and 80.3% (-0.1%) on ImageNet-1K, respectively. This verifies the spatial modeling ability of our V2M.
>
> ## Q3: Classification head
>
> Thanks for this comment. We employ a common form of classification head. Specifically, we first use an MLP to project the feature dimensions to 1000 (the number of classes in the dataset), and then after operations such as log softmax, we constrain it with the ground truth through cross-entropy.
>
> ## Q4: V3M
>
> This is an exciting suggestion. V3M can be considered as a direct extension of V2M in the dimension of state. While V2M performs state inference in both horizontal and vertical directions, V3M, tailored for video data, requires the addition of a temporal dimension for more complex modeling processes. In the computational process, we can also refer to the approach of V2M, simplifying the three-dimensional inference into the form of three one-dimensional SSMs, which facilitates the acceleration of the training process. As for the specific results of V3M on video data, we will delve into this in the future, and we appreciate again for raising this valuable suggestion.

---

> ### Author Response · Authors · 2024-12-02
> **Gentle Reminder Regarding Review of Reviewer RmnR**
>
> Dear reviewer RmnR,
>
> I hope this message finds you well. We greatly appreciate the valuable feedback and suggestions you have provided so far. As the deadline approaches, we are eager to receive your feedback on our response and revisions. If possible, we kindly request an update on the progress of the review to ensure we can address any further comments or revisions promptly.
>
> Should you require any additional information or assistance from our end to help facilitate the review process, please do not hesitate to let us know. Your insights are highly valuable to us, and we genuinely appreciate your time and effort in reviewing our paper.
>
> Thank you for your patience and cooperation. We are looking forward to hearing from you soon.
>
> Warm regards,
>
> Submission2798 Authors.

---

> > ### Comment · Reviewer_RmnR · 2024-12-03
> >
> > Thanks for the authors' response. I keep my original rating.

---

### Meta-Review · Area_Chair_tH7o · 2024-12-19

**Metareview:**

This paper proposes a Mamba-based backbone for vision tasks, using a 2D SSM formulation to effectively incorporate 2D locality priors into vision data. Reviewers generally find the motivation for this approach intuitive and interesting; however, they raised significant concerns regarding the method's complexity and performance. After the rebuttal and discussion phases, some major issues remain unaddressed.

The paper received mixed reviews, with scores of 6, 5, 3, and 8, resulting in an average score of 5.5. The Area Chair carefully reviewed the paper and agreed with Reviewer 7MsE that the limited performance and efficiency could significantly diminish the significance of this method as a Mamba-based vision backbone. Furthermore, the novelty seems limited due to the existence of several prior works that utilize 2D scans. In light of these factors, the Area Chair recommends rejecting the paper.

**Additional Comments On Reviewer Discussion:**

The main concerns raised by reviewers focus on the paper's novelty, implementation, performance, experimental results, and overall presentation. During the rebuttal and discussion phases, some issues, such as presentation and implementation, were adequately addressed. However, reviewers still express ongoing concerns regarding the paper's limited performance and efficiency. Additionally, the aspect of novelty has not been fully clarified, particularly in relation to prior works that utilize 2D scans. As a result, the AC recommends rejecting the paper.

---

### Decision · Program_Chairs · 2025-01-22

Reject